# A Review of Treatment Methods Focusing on Human Induced Pluripotent Stem Cell-Derived Neural Stem/Progenitor Cell Transplantation for Chronic Spinal Cord Injury

**DOI:** 10.3390/medicina59071235

**Published:** 2023-07-01

**Authors:** Takahiro Shibata, Syoichi Tashiro, Masaya Nakamura, Hideyuki Okano, Narihito Nagoshi

**Affiliations:** 1Department of Orthopaedic Surgery, Keio University School of Medicine, 35 Shinanomachi, Shinjuku-ku, Tokyo 160-8582, Japan; 2Department of Physiology, Keio University School of Medicine, 35 Shinanomachi, Shinjuku-ku, Tokyo 160-8582, Japan; 3Department of Rehabilitation Medicine, Keio University School of Medicine, 35 Shinanomachi, Shinjuku-ku, Tokyo 160-8582, Japan

**Keywords:** chronic spinal cord injury, cell transplantation, rehabilitation, combination treatment

## Abstract

Cell transplantation therapy using human induced pluripotent stem cell-derived neural stem/progenitor cells (hiPSC-NS/PCs) has attracted attention as a regenerative therapy for spinal cord injury (SCI), and its efficacy in treating the subacute phase of SCI has been reported in numerous studies. However, few studies have focused on treatment in the chronic phase, which accounts for many patients, suggesting that there are factors that are difficult to overcome in the treatment of chronic SCI. The search for therapeutic strategies that focus on chronic SCI is fraught with challenges, and the combination of different therapies is thought to be the key to a solution. In addition, many issues remain to be addressed, including the investigation of therapeutic approaches for more severe injury models of chronic SCI and the acquisition of practical motor function. This review summarizes the current progress in regenerative therapy for SCI and discusses the prospects for regenerative medicine, particularly in animal models of chronic SCI.

## 1. Introduction

Spinal cord injury (SCI) is a traumatic injury to the parenchyma of the spinal cord that results in severe physical disabilities, including permanent paralysis below the level of injury, sensory loss, and bladder dysfunction. Worldwide, the incidence of SCI is estimated to be 180,000 per year, with an estimated 3 million people suffering from the consequences of SCI. In recent years, remarkable progress in basic research has demonstrated the efficacy of regenerative medicine, such as transplantation of human induced pluripotent stem cell-derived neural stem/progenitor cells (hiPSC-NS/PCs), in many cases for the treatment of subacute SCI [1,2]. Human iPS cells were established from derived somatic cells by Yamanaka et al. at Kyoto University, Japan in 2006 [3]. iPS cells can proliferate without limit and differentiate into any cell type, and hiPSC-NS/PCs have been induced via the neurosphere method using iPS cells [4]. However, few studies have focused on the chronic phase of SCI, which accounts for the majority of SCI patients, suggesting a high treatment hurdle. Although the reconstruction and remyelination of damaged pathways are considered an important strategy for regenerative treatment of SCI using hiPSC-NS/PC transplantation [5], the presence of scarring, muscle contractures, and muscle weakness, which are unique to the chronic phase, may complicate treatment [6,7]. Therefore, the treatment strategy for the chronic phase should include a combination of therapies that have been shown to be effective in the subacute phase [8]. In this review, we summarize our recent studies on the combined treatment of chronic SCI with hiPSC-NS/PC transplantation and rehabilitation, review previous reports on regenerative therapy for the chronic phase, and consider the prospects for therapeutic strategies.

## 2. Time-Dependent Changes in the Microenvironment of the Injured Spinal Cord

Most patients with SCI are already in the chronic phase. However, basic research to date has reported the efficacy of cell transplantation therapy mainly in the subacute phase, whereas cell transplantation therapy alone has not been sufficiently effective in the chronic phase [6,7,9,10]. The environment of the injured spinal cord in the chronic phase is very different from that in the acute to subacute phase, and regeneration is extremely difficult owing to various factors such as scars and cavities.

The pathophysiology of the injured spinal cord changes over time after injury. First, there is mechanical injury (primary injury) due to direct external forces, such as traffic accidents and sports injuries, followed by secondary injury that extends the damage to the injured spinal cord. In this process, the blood–spinal cord barrier is disrupted, causing neuronal cell death due to oxidase induction, increased production of inflammatory cytokines due to immune cell infiltration, and local ischemia and edema due to disruption of the vascular structure. This period of intense inflammation is called the acute phase, which lasts for approximately one week after injury in the rodent model. Subsequently, during the subacute phase (1–2 weeks after injury), reactive astrocytes migrate around the injured area to surround the inflammatory cells infiltrating the injured area and prevent the damage from spreading. Eventually, glial scars and cavities are formed in the injured area, and the patient progresses to the chronic phase (6 weeks after injury), in which spinal cord function is permanently impaired, including increased axonal growth inhibitors [11]. The environment of the injured spinal cord varies greatly depending on the time of injury; therefore, it is necessary to select the appropriate treatment for each stage of the disease.

## 3. Neural Stem/Progenitor Cell Transplantation for Spinal Cord Injury

### 3.1. Optimal Time for Transplantation

The optimal engraftment timing of transplanted cells is an important parameter in determining treatment efficacy. It has been shown that the acute phase is not conducive to the successful engraftment of transplanted cells [11,12] because the strong inflammation in the acute phase, especially the action of complement proteins such as C5a, can damage the transplanted cells and prevent their differentiation into mature neurons [13]. In contrast, cell transplantation alone was found to be ineffective in the chronic phase when microenvironmental changes such as glial scarring formed around the epicenter of the injury [6]. Although the transplanted cells could survive, the strong intercellular adhesion in the glial scar and axonal growth inhibitors released from the scars prevented the therapeutic effect [11]. Therefore, the subacute phase, when the inflammation has subsided but the glial scar has not yet fully formed, is considered the optimal time window for transplantation [12], and most cell transplantation therapy studies—including ours—have mainly focused on the subacute phase.

### 3.2. Transplantation in the Subacute Phase

Cell transplantation therapy using NS/PCs is attracting attention as a regenerative therapy for SCI [14]. The proposed mechanisms of functional recovery by transplantation therapy of NS/PCs include the reconstruction of neural circuits by axon elongation, remyelination of demyelinated axons, and tissue protection by neurotrophic factors secreted by transplanted cells [15]. As mentioned above, we and others have conducted numerous studies on hiPSC-NS/PC transplantation therapy for subacute spinal cord injury and have shown the efficacy of cell transplantation therapy [5,16]. Clinical trials in human patients have also been initiated based on the results of preclinical studies [17]. However, the effect of cell transplantation therapy on chronic SCI is poor, and problems remain [1].

### 3.3. Transplantation in the Chronic Phase

As mentioned above, the microenvironment of the chronically injured spinal cord is very different from that in the subacute phase, and the spinal cord is extremely difficult to regenerate owing to various factors that inhibit axon extension and cavity formation. In particular, scarring is a major problem in cell transplantation therapy for chronic SCI because it prevents the engraftment of transplanted cells. It has been reported that the efficacy of NS/PC transplantation alone for chronic SCI is poor, and reports of cell transplantation therapy for chronic SCI have been limited to the efficacy of transplantation of NS/PCs overexpressing neurotrophin-3 (NT-3) [18].

Recently, the efficacy of hiPSC-NS/PCs pretreated with a gamma-secretase inhibitor (GSI), which activates p38 MAPK to promote axonal regrowth, was investigated in an animal model of chronic SCI [19]. The researchers reported that motor functional recovery was achieved through axon regeneration and elongation, even in the chronic phase environment. This is a significant achievement for the clinical application of hiPSC-NS/PC transplantation in patients with chronic SCI because transplantation treatment may be accompanied by invasion or other adverse effects. However, the degree of improvement in motor function was limited compared to the therapeutic effect in the subacute phase. Therefore, it is necessary to investigate the efficacy of a comprehensive treatment that combines cell transplantation with drug administration and rehabilitation in the future. In particular, rehabilitation is known to promote axon elongation, remyelination, and secretion of neurotrophic factors in the spinal cord, and transplanted cells are thought to have the potential to enhance these effects [20].

## 4. Rehabilitation Therapy for Spinal cord Injury in Preclinical Studies

Rehabilitation for SCI is essential to promote the recovery of motor function, prevent joint and muscle contractures and disuse, regain activities of daily living, and restore social participation. There are numerous reports that rehabilitation improves motor function in animal models of acute and subacute SCI [21,22,23,24,25,26,27,28,29,30]. Although the mechanism is still in the hypothetical stage, it is thought to involve the construction of new neural circuits, selective and stable re-enforcement of the locomotor center, synaptic enhancement with long-term structural changes, correction of abnormal plasticity, integration with sensory input, activation of cortical plasticity, and amelioration of spasticity [20,31]. In addition, restoration of spinal cord inhibition and reduction of abnormal pain sensations have been reported [32]. Neurotrophic factors contribute to these effects, and exercise therapy increases the expression of brain-derived neurotrophic factor (BDNF), NT-3, and insulin-like growth factor-I (IGF-1) and induces functional recovery [33,34,35,36,37,38].

Moreover, particularly in recent years, the concept of neurorehabilitation based on neuroscience has gained popularity, not only for the prevention of muscle strength loss but also for the promotion of plasticity of neurological function after injury, and both basic and clinical research are underway [39,40]. Studies have reported various benefits of neurorehabilitation. Functional electrical stimulation (FES), including neuromuscular electrical stimulation (NMES), is a method of assisting a patient’s impaired biomuscular function through electrical stimulation; NMES assists patients with voluntary movement by enhancing movement intention through well-timed and coordinated electrical stimulation [41]. In preclinical studies, current stimulation has been shown to induce migration and proliferation of neural progenitor cells and the expression of neurotrophic factors; therefore, it is also expected to be more effective when combined with stem cell transplantation [42]. Other neurorehabilitation techniques include body weight-supported treadmill training (BWSTT) and robot-assisted gait training (RAGT). These can improve the gait of neurologically ill patients more easily, quickly, and safely than regular ground-based training [43].

However, there are few reports on animal models of chronic SCI. Shibata et al. established a protocol for treadmill training with progressive intensity based on the overload principle [44]. This rehabilitative training improved hindlimb motor function to some extent in mice with thoracic contusive injury, even during the chronic phase. However, this did not lead to the acquisition of a practical gait, suggesting that further additional treatment is needed.

## 5. Drug Intervention for Scars in Chronic Spinal Cord Injury

The presence of glial and fibrous scars, which are unique to the injured spinal cord in the chronic phase, greatly inhibits the effectiveness of treatment. In addition, several molecules expressed in these scars have been identified as inhibitors of axonal regeneration, including chondroitin sulfate proteoglycan (CSPG) [45] and semaphorin3A [46].

There are many studies on ChABC, an enzyme that degrades CSPG [45,47,48,49]. CSPG is an extracellular matrix molecule that can be generated by all neuronal cell types and is highly upregulated in glial scars after nervous system injury. It is widely recognized that CSPG prevents axonal elongation and cell migration across lesions in chronic SCI. ChABC has been highlighted as an enzyme that effectively degrades CSPG and promotes functional improvement in rodent models when administered intrathecally. Recently, lentiviral ChABC transgenesis of the rat spinal cord with extensive CSPG digestion resulted in reduced cavitation volume and improved axonal preservation [50].

Drugs that inhibit the action of axon regeneration inhibitors alone can be effective, but their use in combination with cell transplantation and rehabilitation has been studied and will be discussed below.

## 6. Combined Therapy for Chronic Spinal Cord Injury

### 6.1. Combination of Cell Transplantation and Rehabilitation

Rehabilitation is a widely feasible treatment in clinical practice because it is versatile and rarely ethically problematic. This concept has been termed regenerative rehabilitation and has been shown to focus on the effects of neuroplasticity from functional training through rehabilitation, increased neurotrophic factor expression from exercise therapy, and conditioning against disuse, all of which can boost the functional recovery effects of cell therapy [39]. However, there are very few reports on the combination of cell transplantation and rehabilitation for SCI [39]. Moreover, most studies used a super-acute transplantation model just after the SCI procedure, which is impractical in clinics despite exerting a greater effect.

Some preclinical studies of combined therapy with cell transplantation and rehabilitation have focused on the subacute to late subacute phase (7 days post injury to 41 days post injury) (Table 1). Keyvan-Fouladi et al. and Ruitenberg et al. combined olfactory ensheathing cell (OEC) transplantation and forepaw reaching rehabilitation in chronic SCI model rats in the 2000s [51,52], but these studies lacked remarkable histological or molecular investigations. Sun et al. implanted both OECs and Schwann cells in moderately contused rats during the subacute period (14 days post injury) and performed bipedal BWSTT. They found that treadmill training significantly contributed to increased serotonergic activity in lumbar enlargement. Motor function was significantly improved in the training group, and the effect was greater in the group that received combined treatment than in the training alone group [53]. Regenerative therapy with OEC transplantation for SCI has begun to be applied in clinical practice, and therapeutic effects are expected [54]. Yoshihara et al. transplanted bone marrow stromal cells at nine days post injury in T9/10 mild thoracic SCI model rats and combined motorized bicycle training [55]. They did not observe any significant recovery in histological assessments. However, a review paper indicated that the dose of intervention (3 days/week) might have contributed to the negative results [39]. Kim et al. showed that the combination of bone marrow stromal cell transplantation and treadmill training in rats with subacute thoracic SCI (seven days post injury) increased BDNF and tropomyosin receptor kinase B (TrkB) expression and improved motor function [56].

To our knowledge, there are only a few studies on the combination of NS/PC transplantation and rehabilitation in SCI, but all of them involved treating the acute to subacute phase, which corresponds to within six weeks of injury [14]. Hwang et al. acutely transplanted NS/PCs into the spinal cord of moderately impaired rats and performed bipedal BWSTT for 8 weeks. The combined treatment significantly improved the survival of transplanted cells, promoted greater differentiation into neurons and oligodendrocytes, and showed greater functional recovery than the control group. It also promoted tissue protection, myelination, and restoration of serotonergic fiber dominance in the lumbar spinal cord. The authors attributed this mechanism to reduced stress from reactive oxygen and nitrogen through insulin-like growth factor (IGF)-1 signaling. They also showed that combined rehabilitation increased the expression of BDNF, GDNF, and NT-3 [33]. Younsi et al. implanted NS/PCs into the injured area of a subacute cervical SCI in rats, followed by six weeks of quadrupedal treadmill training. This rehabilitation increased graft survival and promoted differentiation into neurons and oligodendrocytes. Furthermore, the combined treatment demonstrated better functional recovery, along with improved myelination, regeneration of descending tracts, and tissue preservation [57]. Dugan et al. compared the effect of the same rehabilitation regimen initiated at 5 or 35 days post injury in combination with late subacute GABAergic neural progenitor cell transplantation at 28 days post injury. They also reported the anti-inflammatory effects of intensive training on a quadruped treadmill. They showed that the expression of the anti-inflammatory marker IL-4 in cerebrospinal fluid increased, and the expression of the inflammatory cytokines tumor necrosis factor-α and interleukin-1β decreased [58]. Lu et al. reported that a combination of rehabilitation and NS/PC transplantation initiated at the late subacute phase (28 days post injury) significantly accelerated functional recovery after SCI [59]. Furthermore, improved functional recovery was associated with increased regeneration of host corticospinal axons in the graft after rehabilitation. This study demonstrated that the combination of rehabilitation with NS/PC transplantation played an important and synergistic role in promoting neural plasticity that supports functional recovery after SCI.

Although none had previously been validated in a chronic animal model, Tashiro et al. investigated the efficacy of rehabilitation using NS/PC transplantation and treadmill training in a mouse model of chronic SCI [60]. The results showed that the combination of cell transplantation and treadmill training resulted in a significant recovery of motor function compared to the untreated group. In histological and electrophysiological analyses, the cell transplantation group showed increased excitability of the central pattern generator in the lumbar spinal cord and improved spinal cord conductivity, whereas the training group showed improved proper inhibition and reduced spasticity. In addition to these synergistic effects, the combination therapy group showed increased differentiation of transplanted cells into neurons and increased synapses and new fibers in lumbar enlargement. However, the combination therapy group was not superior to either single-therapy group in motor function evaluation; therefore, it is indispensable to find methods to maximize the effects of cell transplantation and rehabilitation.

### 6.2. Combination of hiPSC-NS/PC Transplantation and Rehabilitation

Considering the conversion of NS/PCs used in preclinical studies into practical cells for clinical applications, it would be useful to use hiPSC-NS/PCs for transplantation. This has great potential for the future development of regenerative medicine, not only because of fewer ethical concerns but also because it preserves the therapeutic properties of plu-ripotent stem cells [4]. This combination is expected to have synergistic effects and may be the best option for the treatment of chronic SCI. However, the therapeutic potential of hiPSC-NS/PCs, especially in combination with rehabilitation, has not been fully eluci-dated. It is essential to have a clear, molecular understanding of whether the combination shows synergistic effects or does not show add-on effects. Therefore, we comprehensively searched for preclinical studies that addressed this combination for SCI; however, only one study fit that criterion [61].

Shibata et al. used immunodeficient NOD-SCID mice with a 70-kilodyne moderate contusion injury to the 10th thoracic spinal cord. Two groups were studied: the combination of hiPSC-NS/PC transplantation and rehabilitation group and hiPSC-NS/PC transplantation monotherapy group. The hiPSC-NS/PCs were transplanted rostrally and caudally to the lesion seven weeks after injury, the chronic phase in rodents. After eight weeks, the efficacy of the combined rehabilitation treatment, applied as quadrupedal treadmill training, was examined by comparing effects on training and histology. A standardized and qualified quadrupedal training protocol was applied [44]. The combination improved survival of the transplanted NS/PCs and promoted their differentiation into mature neurons. Improved survival is one benefit of adding rehabilitation training to transplantation. Regardless of the timing of injury, the survival of grafted cells rapidly declines within two weeks of transplantation; therefore, preventing this decline is crucial for improving motor function. This study showed for the first time that combination rehabilitation prevents the loss of transplanted cells in both the chronic and subacute phases. Differentiation of transplanted cells into neurons and oligodendrocytes is another essential factor for functional recovery. Previous studies focusing on the subacute stage demonstrated that transplanted neural stem cells differentiate into more mature neurons and oligodendrocytes after rehabilitative training. In contrast, the results of the current study were limited to an increase in the differentiation of hiPSC-NS/PCs into mature neurons during rehabilitation, consistent with previous findings that focused on the chronic phase [49]. These findings imply that the timing of the training sessions may alter how the differentiation of grafted neural stem cells is affected by rehabilitation. More neurotrophic factors, such as BDNF and NT-3, were expressed in the spinal cord tissue, including the injured area, and increased neuronal activity and 5-HT-positive fibers were observed in the spinal cord. The raphespinal tract, identified by 5-HT staining, is a supraspinal tract that originates in the brainstem and is crucial for forelimb and hindlimb coordination, contributing to gait quality in quadrupedal rodents. These findings demonstrate that the combined treatment increased numbers of 5-HT fibers in the lumbar spinal cord, especially in the case of chronic SCI. It has been documented that hiPSC-NS/PC transplantation or rehabilitation training alone can boost 5-HT fibers [32,33]. Combining transplantation and rehabilitation improved recovery of motor function more than cell transplantation alone (Figure 1). Motor function was evaluated from multiple perspectives, including general outdoor exercises using a BMS, quadrupedal appearance using the DigiGait system footprint, and joint movement using kinematics captured by a high-resolution camera. Endurance, coordination, and disuse muscle atrophy were evaluated using the rotarod test, phase dispersion analysis, and muscle weight, respectively. When the results of electrophysiological studies were also considered, these parameters were more improved in the combination of the hiPSC-NS/PC transplantation group and the rehabilitation group than in the hiPSC-NS/PC transplantation monotherapy group. Using hiPSC-NS/PCs pretreated with GSI, when combined with rehabilitation, produced a significant improvement in practical motor function; however, a previous report [19] showed that treatment with GSI-pretreated hiPSC-NS/PCs alone also produced some improvement in motor function in the chronic phase. This improvement may have been partly due to the strengthening of local circuit synapses in the lumbar spinal cord and recovery of leg-muscle mass solely through rehabilitation therapy. However, as shown by analyses of synaptic interactions between the transplanted and host cells and electrophysiological analysis, increased synaptic coupling may have been due to improved survival of the transplanted cells and enhanced neuronal differentiation. Thus, the combination of transplantation and rehabilitation may play an essential role in the integration of lower and upper motor neurons. This study was the first report on combining hiPSC-NS/PC transplantation with rehabilitation for chronic spinal cord injury, which is a significant achievement in establishing a therapeutic basis for regenerative medicine for chronic SCI in clinical practice. However, a limitation of this study is that the animal model used was a moderate contusion injury of the thoracic spinal cord; it is necessary to investigate more severe injuries. There remains room for improvements. For example, the combination of hiPSC-NS/PCs and rehabilitation did not reduce scar size, suggesting that further additional treatment intervention to counteract scarring may be necessary in more severe models.

## 7. Combination of Regenerative Therapy and Drugs or Biomaterials

Many reports on combination therapy with pharmacotherapy have shown significant improvements in motor function. Glial and fibrous scars, together with the axonal growth inhibitors expressed therein, are notable inhibitory factors in the chronically injured spinal cord; therefore, they need to be addressed to improve therapeutic efficacy. As mentioned above, there have been numerous reports on combination therapy with ChABC, an enzyme that degrades CSPG—a major component of glial scars that physically inhibits axonal regeneration. It has been reported that the combined administration of ChABC and treadmill training in chronic cervical and thoracic SCI models results in axonal regeneration and recovery of motor function [48,49]. The efficacy of the combined treatment of ChABC and hiPSC-NS/PC transplantation for cervical and thoracic SCI was also confirmed in the chronic phase. Researchers have reported that combination treatment improved the survival of transplanted cells and myelination in the host [62,63].

On the other hand, an axonal growth inhibitor, semaphorin3A, is also attracting attention. In addition to the effect of the drug on promoting axonal growth, the semaphorin3A inhibitor, applied via a sheet-type sustained-release system, induces spinal cord plasticity and motor function recovery in combination with treadmill training in the subacute phase [64]. It is believed that this drug delivery system will be suitable for artificial dura mater used in surgical treatment.

Recently, a combination of regenerative medicine and biomaterials has been investigated. It has been reported that combined treatment with a drug sustained-releasing collagen scaffold containing hepatocyte growth factor (HGF) and hiPSC-NS/PCs resulted in functional improvement even in chronic complete SCI [65]. In this study, the scaffolds showed accelerated angiogenesis, anti-inflammatory effects, and axonal regeneration around the lesions. When combined with hiPSC-NS/PC transplantation, increases in cell viability and axonal projections of both transplanted and host cells were observed. Further combination with rehabilitation aimed at the activation of neural circuits and ChABC leading to scar resolution may induce synergistic functional recovery.

## 8. Conclusions

This paper reviewed the progress of basic SCI research targeting the chronic phase. While there are a variety of treatment options, the appropriate methods are highly dependent on the time since injury and the level and severity of the injury. Considering the reports to date, the combination of cell transplantation, drug therapy including biomaterials, and rehabilitation will have a remarkable therapeutic potential by acting on multiple points, such as the lesion tissue and lumbar enlargement, and could be applied to all cases of chronic SCI.

Thus, there is still a long way to go to overcome chronic SCI, but steady progress is still being made. However, further progress is required.

## Figures and Tables

**Figure 1 medicina-59-01235-f001:**
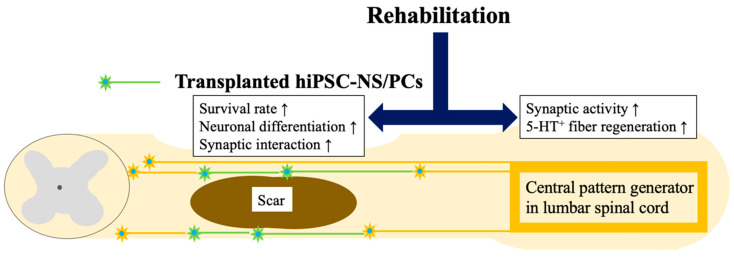
Rehabilitation training was able to enhance the synaptic activity at the central pattern generator in the lumbar spinal cord and promoted survival rate and neural differentiation of grafted hiPSC-NS/PCs. Consequently, the combination therapy with hiPSC-NS/PC transplantation and rehabilitative training has significantly improved motor functions.

**Table 1 medicina-59-01235-t001:** Preclinical studies of combination therapy with cell transplantation and rehabilitation.

Author (Year)	SCI Animal Model	Cell Source	Training Type	Outcomes of Study
Keyvan-Fouladi N (2003)	Late-subacute contusive rats	OECs	Forepaw reaching	Functional repair of corticospinal tracts.
Ruitenberg MJ (2005)	Late-subacute contusive rats	OECs	Forepaw reaching	Increase in the number of corticospinal axons.
Sun T (2013)	Subacute contusive rats	OECs + Schwann cells	Bipedal treadmill training	Enhancement of increased serotonin activity at lumbar enlargement.
Yoshihara H (2006)	Subacute contusive rats	Bone marrow stromal cells	Bicycle training	No difference in cell survival, evidence of axonal growth into grafts and lesion size.
Kim YM (2018)	Subacute contusive rats	Bone marrow stromal cells	Quadrupedal treadmill training	Increase expression of BDNF and TrkB and improvement motor function.
Hwang DH (2014)	Subacute contusive rats	NS/PCs	Quadrupedal treadmill training	Improvement in NS/PC survival and differentiation into neurons and oligodendrocytes. Attenuation in cellular stresses from reactive nitrogen and oxygen via IGF-1 signaling.
Younsi A (2020)	Subacute contusive rats	NS/PCs	Quadrupedal treadmill training	Graft survival, and differentiation into neurons and oligodendrocytes increased. Better functional recovery with synergistic effect.
Dugan EA (2020)	Late-subacute contusive rats	GABAergic -NS/PCs	Quadrupedal treadmill training	Enhancement of neuropathic pain reduction as assessed by allodynia and hyperalgesia. Restoration of GABAergic neuronal and process density.
Lu P (2022)	Late-subacute contusive rats	NS/PCs	Forepaw reaching	Increase host corticospinal axon regeneration into grafts and improvement of motor function.
Tashiro S (2016)	Chronic contusive mice	NS/PCs	Bipedal treadmill training	Facilitation of neuronal differentiation of transplanted cells. Enhancement of neurogenesis in lumbar enlargement.

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
