# Peer review of "A Review of Treatment Methods Focusing on Human Induced Pluripotent Stem Cell-Derived Neural Stem/Progenitor Cell Transplantation for Chronic Spinal Cord Injury"

_medicina, 2023, doi:10.3390/medicina59071235_

Round 1
Reviewer 1 Report
This paper is a review of studies in which stem cells are used to create neurons which are implanted into the spinal cord after injury. It focusses on treatment in the chronic phase and almost all the papers appear to be about rodent studies. I do not work in this field and I found the paper concise and interesting. In brief: to be effective, stem cell treatment must be combined with rehabilitation to promote neuroplasticity; the inhibitor GSI helps, but there are many other molecules that might be beneficial (chondroitin, semaphorin, etc).
The paper could be clearer. I make the following suggestions.
In the Abstract, you say ‘... few studies have focused on the chronic phase ...’. I wondered whether you meant that treatment was applied in the chronic phase, or the chronic outcome of treatment in the acute phase. I suggest you change the sentence to ‘... few studies have focused on treatment in the chronic phase ...’.
hiPSC-NS/PC. For non-specialists, it would help if you gave more explanation of what this means. My guess is: stem cells that have been converted to neurons and surgically-implanted at the lesion. What is it?
Line 36: the chronic phase ‘accounts for 95% of SCI patients’. This is an odd thing to say. Surely a more relevant is that one wants to treat people living with SCI not just those recently injured? Other advantages are: (i) the effect will be clearer when the natural rate of change is small. and (ii) probably easier ethically when patient has time more time to consider risk/benefit of proposed procedure.
Line 40 ‘scarring and muscle contractures’. Why only mention these effects of chronic SCI? What about muscle weakness or spasticity? These are many sequelae.
Section 4. I was uncertain what you were saying here. The sentences at 120 and 133 appear contradictory.
Can you improve the headings of sections 6.1 and 6.2 please. To me they mean the same. Is 6.2 about use in man?
Line 208. There is no Figure 1. Is this S1 (which I have not seen)?
Line 244. ‘4Discussion’
Line 97 ‘extremely difficult to regenerate’. What do you mean? Do you mean that the environment deters natural recovery, or that it inhibits implanted neurons, or that it is difficult to imitate in the lab?
Line 109 ‘limited and insufficient’. What do you mean here? Insufficient for what?
Line 165 ‘and histologic findings secondary to the combined treatment.’ I do not understand what this means.
Reviewer 2 Report
Summary:
This review by Shibata et al. focuses on human iPSC-derived neural stem and progenitor cell treatment for SCI, with a particular emphasis on the development of therapies that are effective to treat chronic SCI. This is an important topic as it is true that most SCI patients are living with chronic injury yet treatment of the chronic population is underdiscussed.
Comments:
· For a broad, unbiased literature review, a reference to a previously published study by the authors within the abstract is inappropriate. The sentence “In our previous study, we found that the combination of validated rehabilitation training and hiPSC-NS/PCs transplantation promoted functional recovery in chronic moderate contusive SCI” seems out of place because it refers to a single study, and should be removed.
· First sentence of Introduction: “Spinal cord injury (SCI) is a traumatic injury to the parenchyma of the spinal cord, and the central nervous system…” The spinal cord is part of the CNS, so “and the central nervous system” is out of place and should be removed.
· The references cited are almost exclusively articles by Japanese research groups. This does not give a well rounded view of the literature. The authors should reference additional studies by groups in other countries (Europe, US, and Canada) to generate a more unbiased literature review.
· In section 2, the authors write “basic research to date has reported the efficacy of cell transplantation therapy mainly in the subacute phase, while cell transplantation therapy alone has not been sufficiently effective in the chronic phase”. Only 1 study is cited, but other students have shown that chronic cell transplantation is possible (especially early studies from the 1980s and 1990s). Please support both of these claims with data.
· Section 3.2: “Cell transplantation therapy using NS/PCs is attracting attention as a new treatment for SCI.” This is not a new treatment – preclinical and clinical NSPC transplantation has been done for decades. Please revise this claim.
Some English language editing is needed.
Reviewer 3 Report
Dear authors. The paper submitted for review has an interesting concept. The paper describes the feasibility of cell therapies depending on the time since injury, the use of biomaterials and the rehabilitation of the patient. The topic is important, but the paper is only an introduction to a serious analysis and review of the available literature. The work should be expanded with the addition of tables. Each of the chapters should be expanded and the proposed mechanisms and actions of the therapy should be described.
In addition, the work should be supplemented with diagrams and a description of how the review was carried out.
Round 2
Reviewer 2 Report
Some of the recommendations from my previous review were adequately addressed, but not all.
My previous recommendation to remove the sentence within the abstract: “In our previous study, we found that the combination of validated rehabilitation training and hiPSC-NS/PCs transplantation promoted functional recovery in chronic moderate contusive SCI” has not been done.
It also looks like my second point regarding the use of “and the central nervous system” has also not been revised.
Regarding the use of “new” to describe cell transplantation therapy, the authors have not sufficiently addressed this concern. For example the first sentence of the abstract calls it a “new” therapy.
It's fine.
Reviewer 3 Report
Thank you for the corrections in your work. Unfortunately, not everything has been included.
1 I kindly ask you to add to the work a table that will synthesize the results described in the work.
2. The description of the doctrine of OEC cells is inaccurate and not up-to-date. I recommend the research of Professor Ying Li of UCL and reading the clinical case of Mr. Dariusz Fidyka.
Round 3
Reviewer 3 Report
Thank you for the corrections in the text of the publication.
Author Response
Thank you for reviewing our manuscript entitled "A Review of Treatment Methods Focusing on hiPSC-NS/PCs Transplantation for Chronic Spinal Cord Injury".
We have had the paper proofread by an English writing expert to improve the quality of the English text. We have added the outcomes of the studies in the table. The key words are listed at the beginning of the paper.
We hope that our revised manuscript meets the standards for Medicina and is now acceptable for publication. We look forward to hearing from you soon. Thank you for your consideration.
